# Different Cropping Patterns to Restore Saline-Alkali Soils in Northeast China Affect the Abundance of Functional Genes in the Soil Nitrogen Cycle

**Junnan Ding [1] , Bin Li [2], Minglong Sun [3] and Xin Li [2],***

1   School of Geography and Tourism, Harbin University, Harbin 150086, China
2   College of Resources and Environment, Northeast Agricultural University, Harbin 150030, China
3   Crop Resources Institute, Heilongjiang Academy of Agricultural Sciences, Harbin 150086, China
*   Correspondence: swx05256lx@126.com; Tel.: +86-15104554929

**Abstract:** Considerable attention has been paid to the establishment of an appropriate cropping patterns for the restoration of saline-alkali lands. This study's goal was to explore changes in nitrogen-cycling (N-cycling) gene (nitrogen fixation: *nifH*; nitrification: AOA, AOB, and *nxrB*; denitrification: *narG*, *norB*, and *nosZ*) abundance of three cropping patterns at two soil depths in saline-alkali soils. Results showed that rotation and mixture promoted soil nutrients. N-cycling functional genes were significantly influenced by soil depths and cropping patterns. Compared with monoculture, rotation decreased the abundance of *nifH*, AOA, *narG*, and *nosZ* and increased the abundance of AOB; mixture decreased the abundance of AOA, *narG*, and *nosZ* and increased the abundance of AOB and *nxrB* in the 0–15 cm soil depth. Rotation increased all genes abundance; mixture increased *nosZ* abundance and decreased *nxrB* abundance in 15–30 cm soil depth. Soil protease, cellulase, nitrate reductase, pH, AK (available potassium), and AP (available phosphorus) were important factors influencing N-cycling gene abundance. In conclusion, rotation and mixture not only reduced soil salinity but also improved soil fertility and nitrogen cycling. These findings can provide some theories for the sustainable development of N-cycling during the restoration of saline-alkali soils.

**Keywords:** nitrogen fixation; nitrification; denitrification; cropping patterns; qPCR

## 1. Introduction

Saline-alkali soils contain high levels of soluble salts. Globally, about 932 million hectares (M ha), including 831 million hectares of agricultural land, are affected by salt damage [1]. Saline-alkaline soils have little productivity [2]. Several nations and regions pay significant attention to the improvement and extension of saline-alkali wastelands, especially considering the present global food crisis, and China has $99.13 \times 10^6$ ha of various forms of saline-alkali land [3]. Natural and anthropogenic increases in soil salinity are a key environmental issue that hinders plant development and production [4–8]. Due to their high salt concentration, saline-alkali soils affect the osmotic potential outside the root system, preventing plants from absorbing water and so affecting plant development [9–13]. The microbial populations responsible for the cycle of organic compounds are under a rising challenge from soil salinity [14]. In order to ensure national food and ecological security, the improvement and usage of salt-affected land are essential.

Planting alkali-tolerant plants has been observed to enhance microbial activity and restore soil fertility [15–17]. Oats, alfalfa, and tall wheatgrass are commonly used as species for the phytoremediation of saline lands. Alfalfa is a $C_3$ plant and one of the world's most important perennial leguminous forage grasses that improve soil fertility [18,19]. In addition, Li et al. [20] found that the introduction of oats mixed with other salinity-tolerant plants significantly contributed to the soil fungal communities in saline-alkali soils in Northeast China. Likewise, tall wheatgrass has a very high salinity tolerance

and is a good choice when considering alkali-tolerant plants [21,22]. It can tolerate salt and sodium stress and has good production [23]. Phytoremediation does not only refer to the cropping of resistant plants but also includes field management practices, such as cropping patterns. Soil rotation has many advantageous uses, such as human food, bioenergy, and animal feed [24]. Crop rotation may exert beneficial effects on the physical, chemical, and biological properties of soils, help in the control of weeds and insect pests, and alleviate the adverse effects of climate and exogenous factors on soils [25]. In addition, mixed cropping systems can leverage resources, enhance biodiversity [26], provide insights for managing agricultural systems more sustainably, and improve ecosystem stability [27–29]. Nevertheless, information on the effects of saline land improvement under different cropping patterns is still insufficient.

Soil microorganisms are often considered indicators of soil health because they show great sensitivity to changes in agroecosystems [30], which is what allows researchers to infer the impact of a process or decision on the soil environment [31]. In addition, the critical contributions of soil biota to ecosystem services have now been universally recognized. It is an important cutting-edge research field of agroecosystems to fully improve the positive influences of soil microorganisms on sustainable agriculture cropping patterns [32]. Microorganisms play a crucial role in soil N-cycling and controlling the amount of soil N accessible to plants [33]. Nitrogen-cycle processes, such as assimilation, ammonification, nitrification, and denitrification, involve microorganisms [34]. Biological nitrogen fixation (BNF) is the primary route for nitrogen intake in the soil-nitrogen cycle, converting molecular nitrogen to ammonia or other nitrogen-containing compounds by the catalytic activity of nitrogenase in nitrogen-fixing bacteria [35]. The critical and rate-limiting phase in autotrophic nitrification is the oxidation of ammonium ($NH_4^+$) to nitrite ($NO_2^-$) by ammonia-oxidizing bacteria (AOB) and archaea (AOA) [36,37]. Nitrite is then oxidized to nitrate ($NO_3^-$), which is gradually reduced by denitrifying bacteria to gaseous products (NO, $N_2O$ and $N_2$), removing nitrogen from the ecosystem [38]. However, information on how agricultural practices change the functional bacteria involved in the nitrogen cycle is still insufficient [39]. Numerous studies have demonstrated that quantitative marker genes are good indicators of key biogeochemical processes in the nitrogen cycle and have been used to quantify the quantity of bacteria in soils using genetic markers [40–42]. In the nitrogen cycle, the *nifH* gene (encoding nitrogenase) is related to nitrogen-fixation; amoA (ammonia monooxygenase) and *nxrB* (Nitrite oxidoreductase enzyme) are related to nitrification; and *narG* (nitric oxide reductase), *norB* (nitric oxide reductase), and *nosZ* (nitrous oxide reductase) are related to denitrification [43].

So far, the impact of different cropping patterns and soil depths on the abundance of functional genes for N-cycling in the saline-alkaline soils of Northeast China is unknown. Therefore, we used monoculture, rotation, and mixture in three experimental field distributions in Northeast China and evaluated the variations in N-cycling gene abundance by qPCR methods. The purpose of our study was to determine if the abundance of N-cycling genes in soil was responsive to changes in cropping patterns and soil depths. We hypothesized that (1) compared to monoculture, rotation and mixture will result in the restoration of soil nutrients, such as soil properties and enzymes; (2) rotation and mixture will cause changes in N-cycling gene abundance; and (3) variations in the abundance of N-cycling genes are associated with soil chemistry and enzymes.

## 2. Materials and Methods

### 2.1. Site Description

The Sifang Mountain Farm is located in Zhaodong City, Heilongjiang Province (125°45′–126°30′ E, 46°12′–46°22′ N), with flat terrain and a single type of landform. The area of soil is carbonate meadow alkaline soil and carbonate meadow soil. The average annual temperature is 2.4 °C, the annual evaporation is 1662 mm, the average annual rainfall is 396 mm, the maximum temperature is 39.0 °C, the minimum temperature is −37.5 °C, and the annual accumulated temperature is 2500–2700 °C. Before restoration, the

soil pH in this location reached 11.00. According to the USDA soil taxonomy system, the soil was predominantly loamy [44] and saline-alkaline experimental, which is a soda saline soil dominated by sodium carbonate bicarbonate.

### 2.2. Experimental Design

This research was conducted in three experimental agricultural plots. These plots consisted of (a) a monoculture system in which alfalfa was cultivated annually; (b) a 13-year-old cycle of alfalfa–oats–tall wheatgrass, with annual alfalfa in the first year, followed by oats in the second year, and tall wheatgrass in the third year, in a three-year rotation; and (c) a mixture system plot in which alfalfa, oats, and tall wheatgrass were simultaneously planted, with each crop planted densely and virtually alternating in every other row. These three cropping patterns use a common approach to fertiliser management, applying nitrogen, phosphorus, and potash in the form of urea, calcium superphosphate, and potassium nitrate. During the experimental period, all plots had similar soil and climate conditions and were fertilized and managed in the same manner.

### 2.3. Soil Sampling

On the 17th of August 2017, soil samples were obtained from three alfalfa-planted plots. Before the alfalfa harvesting, samples were taken. A soil auger was used to collect soil samples from 0–15 cm and 15–30 cm depth in each repeated plot (8 cm diameter and 15 cm deep). Five samples were gathered in an "S" formation and then thoroughly combined to produce one composite sample for each replication, resulting in a total of 18 samples at the plot level. After removing roots and debris, each composite field soil sample was homogenized, packaged in sterile plastic bags, and delivered to the laboratory immediately. The soil was then passed through a 2 mm sieve and each soil sample was divided into two parts. One portion was frozen at $-80\ °C$ for DNA extraction, while the other was air-dried at ambient temperature for soil chemical testing.

### 2.4. Soil Chemical Analysis

The examined soil properties, including soil pH, exchange capacity (EC), soil organic matter (SOM), soil total nitrogen (TN), soil total phosphorus (TP) and potassium (TK), and soil available phosphorus (AP) and potassium (AK), were described by previous studies [45,46]. According to the description of Guan et al. [47], we determined the soil-enzyme activity. The urease activity was determined using urea as a substrate. The protease activity was determined using casein as a substrate. The nitrate reductase activity was determined using nitrate of potash as a substrate. The cellulase activity was determined using carboxymethylcellulose as a substrate. The β-glucosidase activity was determined using β-glucoso-saligenin (salicin) as a substrate.

### 2.5. Soil DNA Extraction and Quantitative PCR (qPCR)

The Power Soil DNA Isolation kit (MO BIO Laboratories Inc., Carlsbad, CA, USA) was used for DNA extraction from 0.25 g of soil following the manufacturer's instructions. The quantity and quality of the extracted DNA were assessed using a NanoDrop ND2000c spectrophotometer (NanoDrop Technologies, Wilmington, DE, USA) and checked on 1% agarose gel. The DNA samples were used to quantify the functional N-cycling genes in the soil samples. Gene encoding the nitrogen fixation (*nifH*), nitrification (*amoA* bacteria, AOB; *amoA* archaea AOA; *nxrB*), and denitrification (*narG*, *norB*, and *nosZ*) processes (Table S1) were performed by quantitative real-time PCR (qPCR). ABI7500 Fluorescent Real-Time PCR Detection System (Applied Biosystems, Carlsbad, CA, USA) for qPCR. Primer sets and the PCR conditions of each gene are detailed in Table S2. The standard curves all had $R^2$ values above 0.99, with slope values in the range of $-3.28$ and $-3.59$ and an estimated amplification efficiency in the range of 101.93 and 89.93%.



*2.6. Statistical Analysis*

The Pearson correlation heat maps of soil properties and N-cycling genes were plotted with the "pheatmap" package [48]. In addition, the different N-cycling gene abundances were ordinated using non-metric multidimensional scaling (NMDS) with the dissimilarity matrices using the "metaMDS" function in the "vegan" package [49]. To analyse the soil chemical drivers of N-cycling gene abundance, we used the "rfPermute" package for random forest analysis [50]. The correlations between N-cycling gene abundance and soil important factors were analysed using Pearson linear regressions.

Using a two-way ANOVA, the impacts of cropping patterns and soil depths on soil chemical properties and enzyme activity were investigated. Subsequently, the effect of cropping patterns on soil properties, enzyme activity, N-cycle gene abundance, and their ratios at two soil depths was investigated using a one-way ANOVA. Before the ANOVA, all data were examined using Levene's test for normality and homogeneity. Differences between groups were examined with Duncan's post hoc test, $p < 0.05$. All the analyses above were carried out in R (v.4.2.2) [51].

## 3. Results

*3.1. Soil Chemical Properties and Soil Enzymes*

The presence of cropping patterns and soil depths can influence soil properties and soil enzymes differently, as shown in Table 1. Results from a two-way ANOVA indicate that pH and TN are significantly influenced by cropping patterns and soil depths, respectively (Table 2). The soil pH in all three cropping patterns ranged from 7.72 to 8.26, indicating that all samples are weakly basic and nearly neutral. Notably, the pH of monoculture soil was higher than that of rotation and mixture soils at two soil depths ($p < 0.05$) (Table 1). With increasing soil depths, the TN content of monoculture, rotation, and mixture soils decreased by 22.99%, 10.79%, and 44.15%, respectively ($p < 0.05$) (Table 1). Cropping patterns and soil depths also significantly influenced EC and AK content ($p < 0.05$) (Table 2). Specifically, at 0-15 cm soil depth, the EC of monoculture soil was significantly higher than that of rotation and mixture soils ($p < 0.05$) (Table 1). In addition, the EC of rotation soil was significantly higher than that of mixture soils ($p < 0.05$) (Table 1). At 15–30 cm soil depth, there were no significant differences in EC among the three cropping patterns ($p > 0.05$) (Table 1). The AK content of mixture soil was significantly higher than that of rotation (+73.22%) and monoculture (+43.67%) soils at 0–15 cm soil depth ($p < 0.05$) (Table 1). At 15–30 cm soil depth, the AK content of rotation soil was significantly higher than that of mixture and monoculture soils ($p < 0.05$) (Table 1).

**Table 1.** The soil chemical properties and soil enzymes in the three cropping patterns.

| | 0–15 cm | | | 15–30 cm | | |
|---|---|---|---|---|---|---|
| | **Monoculture** | **Rotation** | **Mixture** | **Monoculture** | **Rotation** | **Mixture** |
| pH | $8.19 \pm 0.08$ a | $7.98 \pm 0.14$ a | $7.72 \pm 0.11$ b | $8.26 \pm 0.13$ a | $8.15 \pm 0.06$ a | $7.73 \pm 0.20$ b |
| SOM ($g \cdot kg^{-1}$) | $49.02 \pm 1.87$ b | $42.05 \pm 8.85$ ab | $55.44 \pm 1.06$ a | $48.78 \pm 2.44$ a | $50.95 \pm 14.84$ a | $46.81 \pm 2.14$ a |
| TP ($g \cdot kg^{-1}$) | $0.54 \pm 0.21$ a | $0.37 \pm 0.09$ a | $0.73 \pm 0.35$ a | $0.39 \pm 0.00$ a | $0.43 \pm 0.05$ a | $0.31 \pm 0.33$ b |
| TN ($g \cdot kg^{-1}$) | $2.74 \pm 0.99$ a | $2.41 \pm 0.24$ a | $3.42 \pm 0.20$ a | $2.11 \pm 0.40$ a | $2.15 \pm 0.17$ a | $1.91 \pm 0.15$ a |
| AP ($mg \cdot kg^{-1}$) | $61.95 \pm 11.53$ a | $52.31 \pm 14.57$ a | $69.24 \pm 29.00$ a | $49.16 \pm 2.07$ b | $58.57 \pm 0.93$ a | $39.31 \pm 4.35$ c |
| AK ($mg \cdot kg^{-1}$) | $50.01 \pm 4.23$ b | $41.48 \pm 6.05$ b | $71.85 \pm 8.80$ a | $43.61 \pm 0.04$ b | $51.61 \pm 3.20$ a | $40.95 \pm 0.92$ b |
| EC ($mS \cdot cm^{-1}$) | $417.84 \pm 1.95$ a | $392.63 \pm 3.03$ b | $382.32 \pm 2.49$ c | $443.23 \pm 12.80$ a | $440.75 \pm 20.31$ a | $435.98 \pm 6.47$ a |
| Urease ($mg \cdot g^{-1} \cdot d^{-1}$) | $0.09 \pm 0.00$ c | $0.12 \pm 0.01$ b | $0.17 \pm 0.01$ a | $0.09 \pm 0.00$ c | $0.10 \pm 0.00$ ab | $0.10 \pm 0.01$ a |
| Nitrate reductase ($mg \cdot g^{-1} \cdot d^{-1}$) | $0.13 \pm 0.01$ b | $0.12 \pm 0.00$ b | $0.17 \pm 0.01$ a | $0.11 \pm 0.02$ b | $0.09 \pm 0.01$ c | $0.13 \pm 0.01$ a |
| Cellulase ($\mu g \cdot 10 \ g^{-1} \cdot d^{-1}$) | $89.91 \pm 0.05$ c | $188.81 \pm 0.02$ b | $216.45 \pm 0.02$ a | $53.28 \pm 0.01$ c | $69.93 \pm 0.01$ b | $136.53 \pm 0.02$ a |
| β-glucosidase ($\mu g \cdot g^{-1} \cdot h^{-1}$) | $34.37 \pm 2.82$ b | $58.03 \pm 6.32$ a | $66.75 \pm 5.26$ a | $23.82 \pm 0.72$ b | $24.20 \pm 1.61$ b | $30.71 \pm 3.09$ a |
| Protease ($\mu g \cdot g^{-1} \cdot h^{-1}$) | $511.86 \pm 98.68$ b | $848.40 \pm 87.25$ ab | $1021.47 \pm 278.07$ a | $399.68 \pm 251.54$ a | $502.24 \pm 64.02$ a | $684.94 \pm 187.52$ a |

Values represented mean $\pm$ standard deviations (n = 3). Different letters stand for significant effects ($p < 0.05$). SOM, soil organic matter; TN, total nitrogen; TP, total phosphorus; AP, available phosphorus; AK, available potassium; EC, electrical conductivity.

**Table 2.** Two-way ANOVAs for the impact of cropping patterns (CP), soil depths (SD) and their interaction (CP × SD) on soil chemical properties and soil enzymes.

|  | CP | | SD | | CP × SD | |
|---|---|---|---|---|---|---|
|  | *F* | *p* | *F* | *p* | *F* | *p* |
| pH | 23.345 | <0.001 | 1.813 | 0.203 | 0.656 | 0.537 |
| SOM | 0.614 | 0.557 | <0.001 | 0.988 | 2.204 | 0.153 |
| TP | 0.692 | 0.519 | 4.342 | 0.059 | 2.846 | 0.091 |
| TN | 1.072 | 0.373 | 13.346 | 0.003 | 2.903 | 0.094 |
| AP | 0.015 | 0.985 | 3.294 | 0.095 | 2.436 | 0.129 |
| AK | 7.935 | 0.006 | 15.482 | 0.002 | 26.804 | <0.001 |
| EC | 6.647 | 0.011 | 76.099 | <0.001 | 3.169 | 0.079 |
| Urease | 74.454 | <0.001 | 71.61 | <0.001 | 32.763 | <0.001 |
| Nitrate reductase | 64.508 | <0.001 | 56.702 | <0.001 | 2.419 | 0.131 |
| Cellulase | 204.903 | <0.001 | 348.367 | <0.001 | 32.880 | <0.001 |
| β-glucosidase | 39.943 | <0.001 | 219.684 | <0.001 | 20.329 | <0.001 |
| Protease | 7.235 | 0.009 | 9.612 | 0.009 | 0.8 | 0.472 |

Table 2 shows that soil enzyme activity is influenced by cropping patterns and soil depths. When comparing different cultivation methods, the activity of urease, cellulase, and nitrate reductase was significantly higher in mixed soils at a soil depth of 0–15 cm, compared to rotated and monoculture soils ($p < 0.05$) (Table 1). Furthermore, the activity of urease and cellulase was significantly higher in rotation soils than in monoculture soils ($p < 0.05$) (Table 1). The activity of β-glucosidase was significantly lower in monoculture soils than in mixed and rotation soils ($p < 0.05$) (Table 1), while the activity of protease was significantly lower in monoculture soils compared to mixed soils ($p < 0.05$) (Table 1). At a soil depth of 15–30 cm, the activity of urease was significantly lower in monoculture soils than in mixed soils ($p < 0.05$) (Table 1). Moreover, the activity of nitrate reductase, β-glucosidase, and cellulase was significantly higher in mixed soils than in rotated and monoculture soils ($p < 0.05$) (Table 1). The nitrate reductase activity was significantly higher in monoculture soils than in rotation soils, while cellulase activity was higher in rotation soils compared to monoculture soils ($p < 0.05$) (Table 1).

### 3.2. N-Cycling Gene Abundance

Cropping patterns and soil depths had a significant impact on all N-cycling functional genes (Table 3). At a soil depth of 0–15 cm, the abundance of *nifH* was significantly higher in monoculture than in rotation ($p < 0.05$) (Figure 1). Similarly, the abundance of AOA, *narG*, and *nosZ* was significantly higher in monoculture than in rotation and mixture soils ($p < 0.05$), while the AOB abundance was significantly lower in monoculture than in rotation and mixture soils ($p < 0.05$). The AOA abundance in mixture soils was significantly lower than in rotation ($p < 0.05$), and the *nxrB* abundance in monoculture and rotation was significantly lower than in mixture soils ($p < 0.05$). There was no significant difference in *norB* abundance among the three cropping patterns ($p > 0.05$), as shown in Figure 1.

**Table 3.** Two-way ANOVAs for the impact of cropping patterns (CP), soil depths (SD) and their interaction (CP × SD) on N-cycling gene abundance.

|  | CP | | SD | | CP × SD | |
|---|---|---|---|---|---|---|
|  | *F* | *p* | *F* | *p* | *F* | *p* |
| *nifH* | 2.34 | 0.014 | 47.58 | <0.001 | 8.46 | 0.01 |
| AOA | 18,587.73 | <0.001 | 23,284.08 | <0.001 | 19,630.12 | <0.001 |
| AOB | 31.88 | <0.001 | 247.85 | <0.001 | 24.57 | <0.001 |
| *nxrB* | 74.17 | <0.001 | 339.44 | <0.001 | 82.10 | <0.001 |
| *narG* | 8.39 | <0.001 | 68.31 | <0.001 | 17.38 | <0.001 |
| *norB* | 9.67 | 0.03 | 76.96 | <0.001 | 1.04 | 0.39 |
| *nosZ* | 29.90 | <0.001 | 200.08 | <0.001 | 29.29 | <0.001 |

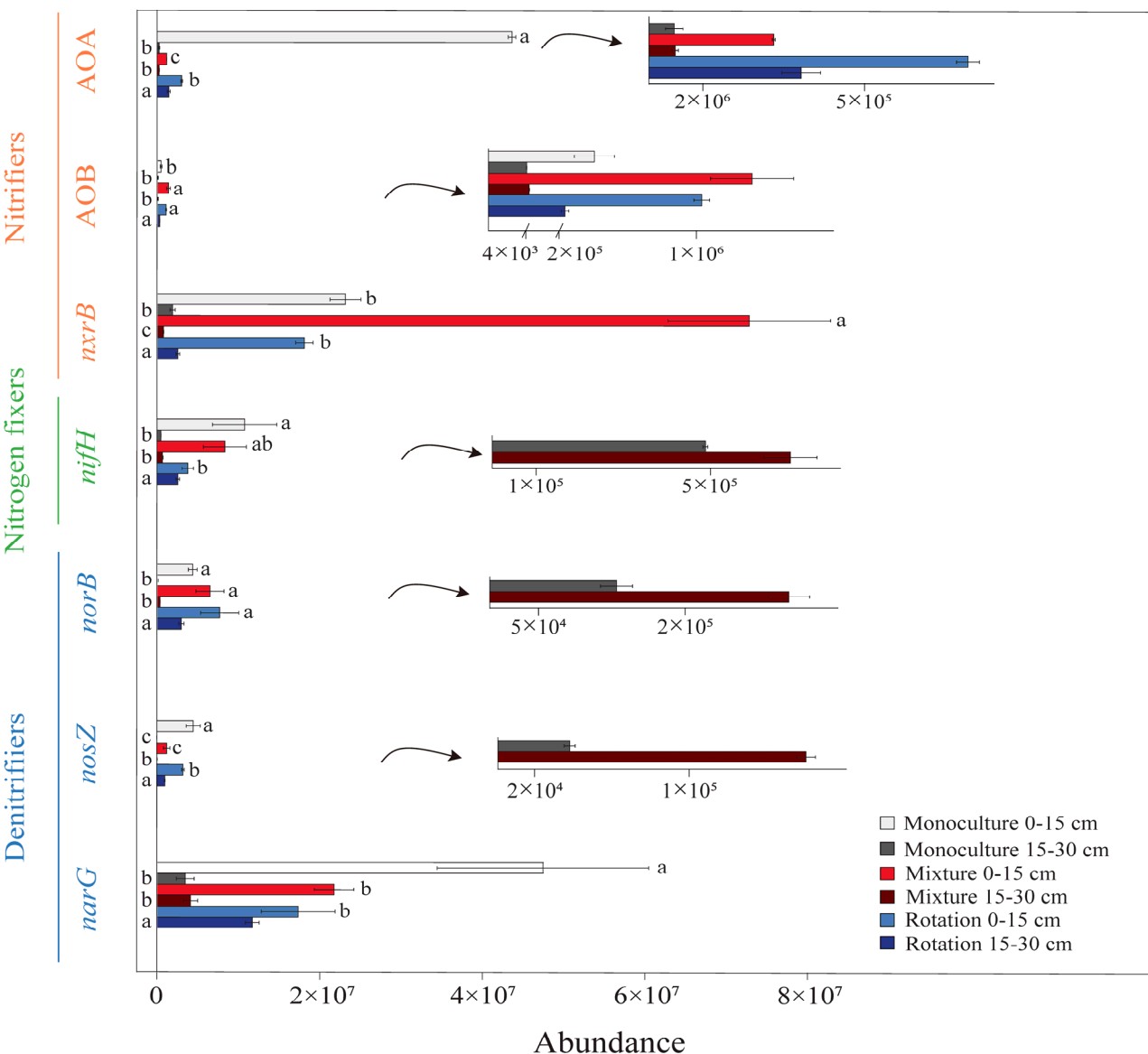

**Figure 1.** The abundance of N-cycling functional genes in different cropping patterns at two soil depths. Different small letters indicate significant differences between treatments ($p < 0.05$). The letters on the right side of the bar chart are for 0–15 cm soil depth, while the letters on the left side are for 15–30 cm soil depth.

At a soil depth of 15–30 cm, all N-cycling abundances in rotation were significantly higher than in mixture and monoculture soils ($p < 0.05$) (Figure 1). In addition, the *nosZ* abundance in mixture soils was significantly higher than in monoculture soils ($p < 0.05$), while the *nxrB* abundance in monoculture was significantly higher than in mixture soils ($p < 0.05$).

### 3.3. N-Cycling Gene Abundance Ratio

To investigate how soil depths and cropping patterns impact N-cycling processes associated with gene abundance, we calculated the ratios for different N-cycling abundances. At surface soil, we found no significant difference in the *nifH*/(AOA +AOB + *nxrB*) ratio among all cropping patterns. However, at deep soil, the *nifH*/(AOA + AOB + *nxrB*) ratio in monoculture was significantly lower than in rotation and mixture ($p < 0.05$) (Figure 2a). The *nifH*/(AOA + AOB + *nxrB*) ratio in rotation and mixture at deep soil was significantly higher than that at surface soil ($p < 0.05$) (Figure 2a). Furthermore, the AOA/AOB ratio

in mixture and rotation was significantly lower than in monoculture at both soil depths ($p < 0.05$) (Figure 2b). The AOA/AOB ratio in monoculture at surface soil was significantly higher than that at deep soil ($P<0.05$) (Figure 2b). We found no significant difference in the *nifH/(narG + norB +nosZ)* ratio among the three cropping patterns at both soil depths ($p < 0.05$) (Figure 2c). The *norB/nosZ* ratio in mixtures was significantly higher than in monoculture at surface soil and lower than in monoculture and rotation at deep soil ($p < 0.05$) (Figure 2d).

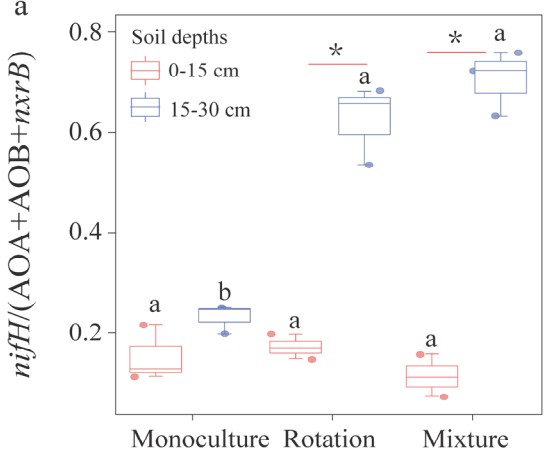
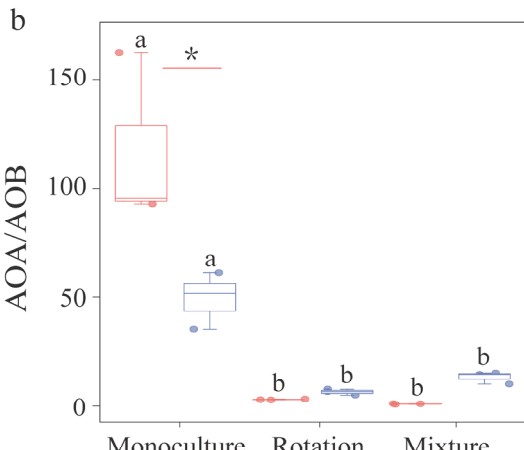
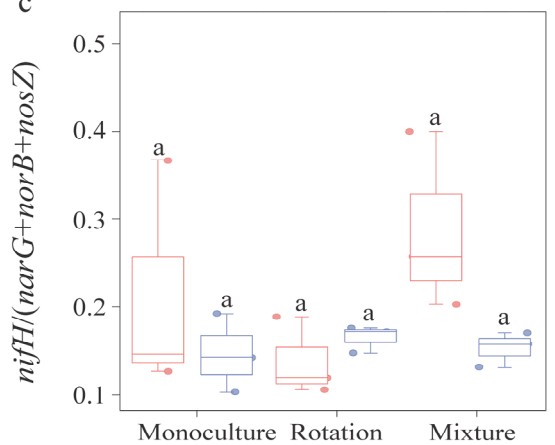
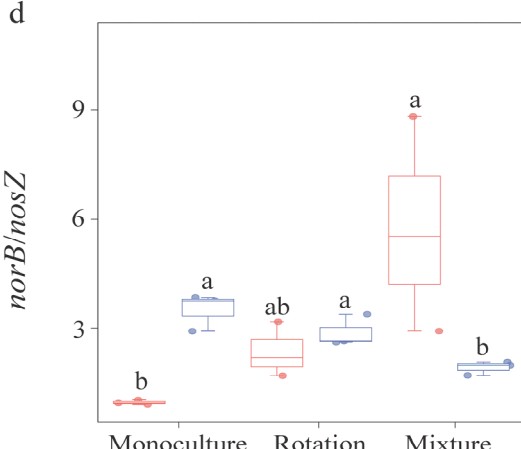

**Figure 2.** The efficiency of N-cycling processes were inferred from the calculation of different N-cycling gene abundance ratios. (**a–d**) represent the nitrogen sequestration process, ammonia oxidation process, nitrogen loss process, and N$_2$O production process, respectively. Different small letters indicate significant differences among the three cropping patterns ($p < 0.05$). A significant difference between soil depths and statistical significance is indicated as follows. * $p < 0.05$.

### 3.4. Association of Soil Properties and Enzyme Activity with Gene Abundance

Based on the NMDS results, we observed that both soil depths and cropping patterns had a significant impact on the abundance of N-cycling genes (Figure 3a). We then proceeded to investigate the individual effects of surface soil and deep soil cropping patterns on the abundance of N-cycling genes (Figure 3b,c). Furthermore, we carried out a correlation analysis between soil properties, enzymes, and N-cycling gene abundance

(Figure 3d,e). Specifically, we found a significant positive correlation between pH and *nosZ* abundance, and a negative correlation between urease and *nosZ* abundance, within a soil depth range of 0–15 cm (Figure 3d). AOB abundance was positively correlated with protease and β-glucosidase, and negatively correlated with soil EC (Figure 3d). Additionally, the *nifH*/(AOA + AOB + *nxrB*) ratio was negatively associated with cellulase, while the *nifH*/(*narG* + *norB* + *nosZ*) ratio was positively correlated with SOM and TP (Figure 3d). Furthermore, the abundance of *nxrB* was positively correlated with TP in deep soil (Figure 3e), and the *norB*/*nosZ* ratio was positively correlated with EC and negatively correlated with protease (Figure 3e).

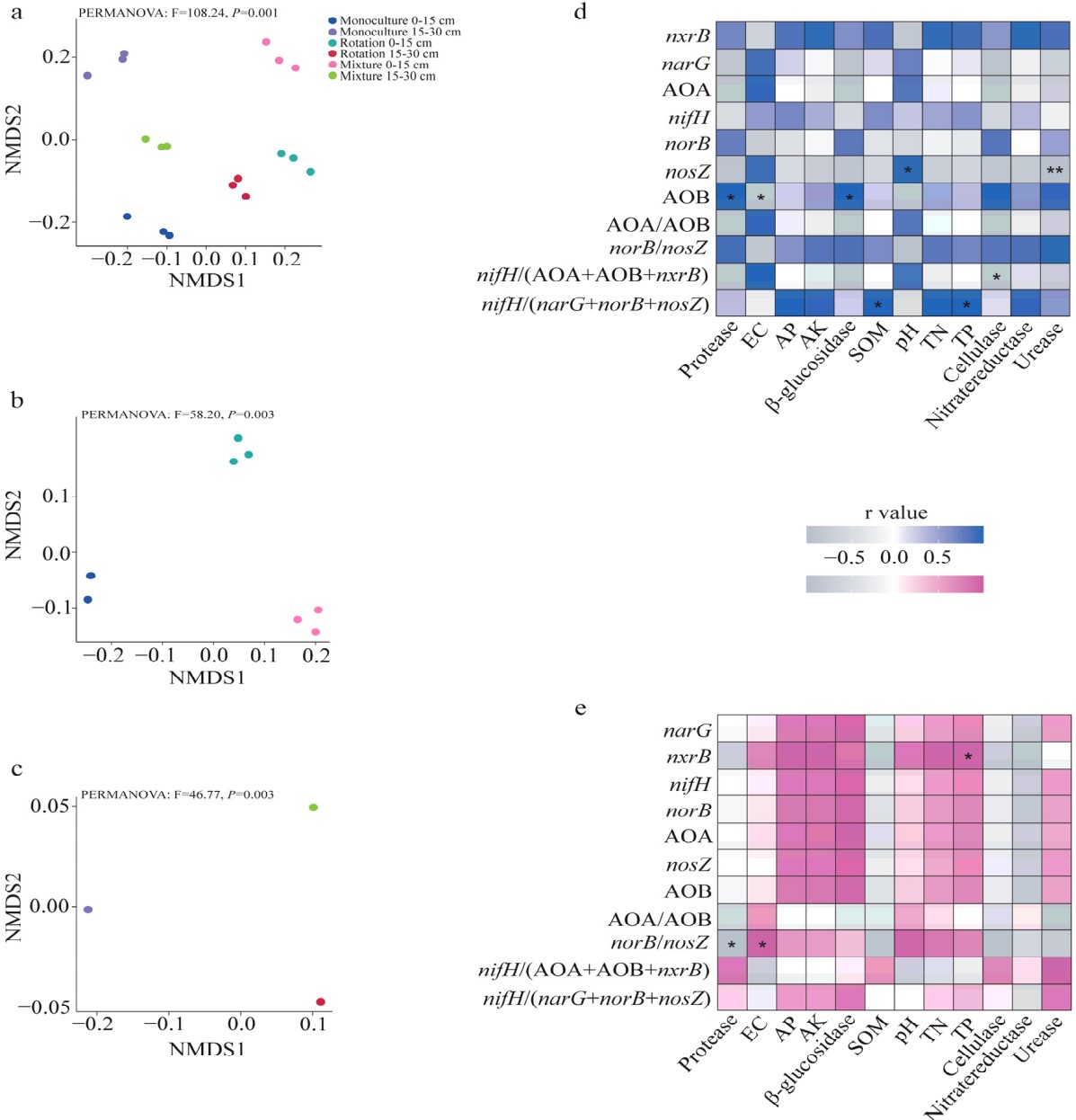

**Figure 3.** The NMDS analysis of cropping patterns and soil depths on functional genes for N-cycling (**a**) and separate analysis of cropping patterns at 0–15 cm (**b**) and 15-30 cm (**c**). Correlation analysis of soil properties and enzyme activities with N-cycling gene abundance at 0–15 cm (**d**) and 15–30 cm (**e**) soil depths. A significant difference between soil depths and statistical significance is indicated as follows. * $p < 0.05$, ** $p < 0.01$.

### 3.5. Random Forest Analysis

We utilized a random forest model (Figure 4) to predict the impact of soil chemical properties and enzymes on the abundance of different N-cycling genes. At a soil depth of 0–15 cm, the abundance of *nifH* and *norB* was primarily driven by the activity of protease, while cellulase was the main driver for the abundance of AOA and *nosZ*. Additionally, pH was the primary driver for the abundance of AOB and *narG*, while nitrate reductase drove the abundance of *nxrB* (Figure 4a). At a soil depth of 15–30 cm, nitrate reductase was the primary driver for the abundance of AOA and *nifH*, AK was the main driver for the abundance of *nxrB*, and AP was the primary driver for the abundance of AOB, *narG*, *norB*, and *nosZ* (Figure 4b). We further investigated the relationship between important soil factors and N-cycling gene abundance through linear regression (Figure 5).

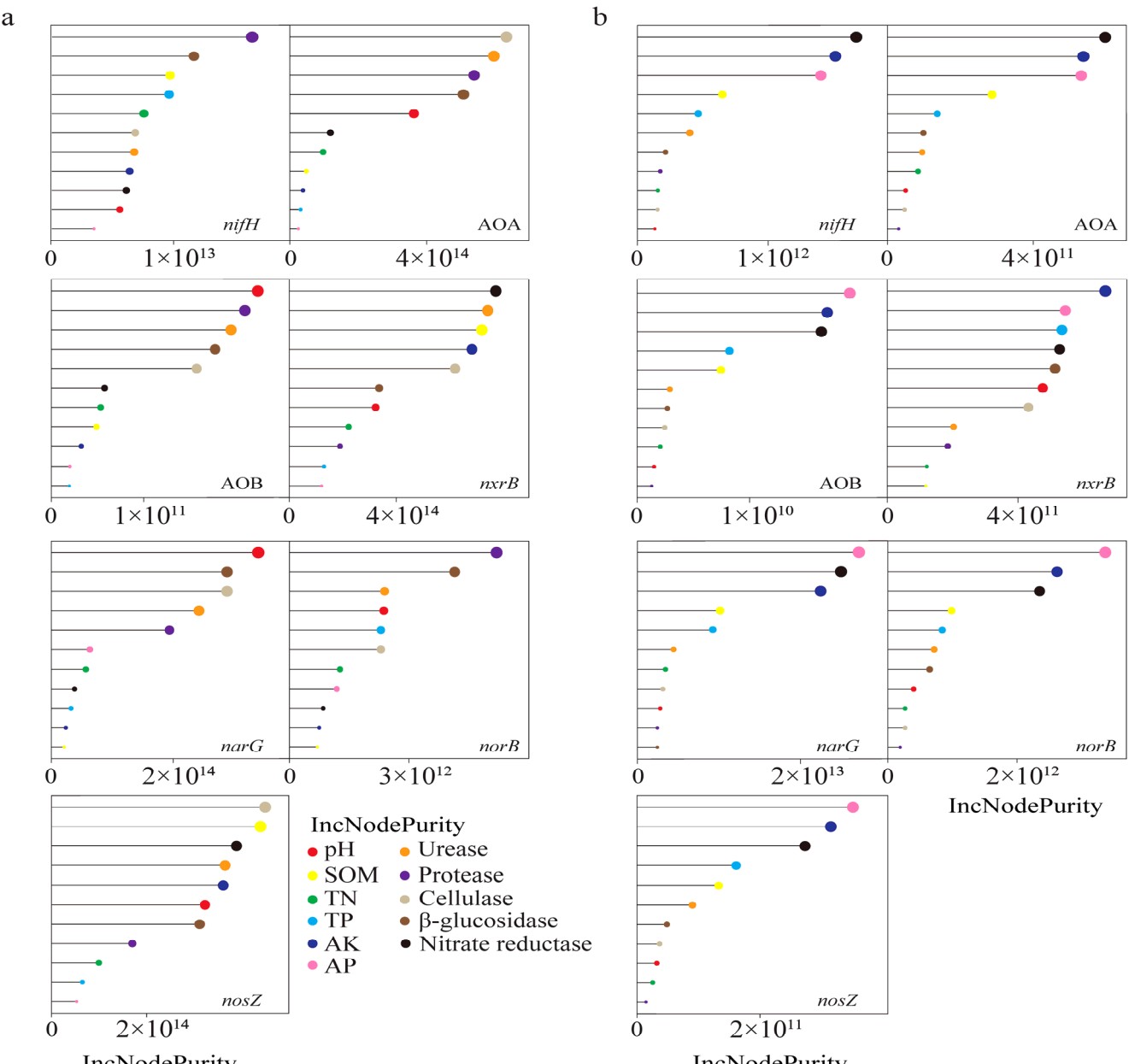

**Figure 4.** Random forest analysis of soil chemistry and soil enzyme activity on functional genes abundance of N-cycling at 0–15 cm (**a**) and 15–30 cm soil depths (**b**).

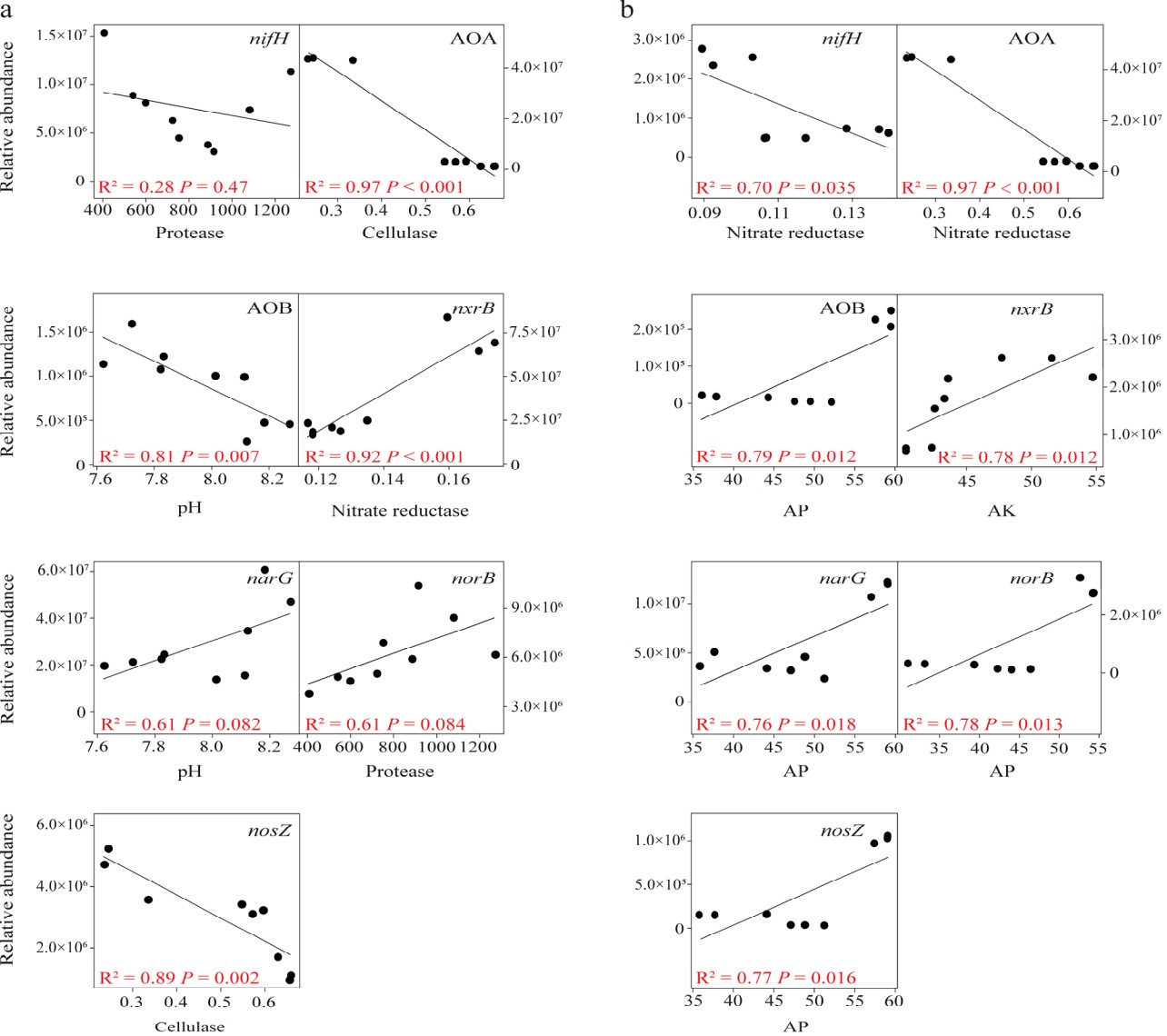

**Figure 5.** Linear regression analysis of important soil properties and soil enzyme on functional genes abundance of N-cycling at 0–15 cm (**a**) and 15–30 cm soil depths (**b**).

A significant linear pattern was observed between soil factors and the abundance of nitrification functional genes (AOA, AOB, and *nxrB*) and *nosZ* at 0–15 cm soil depth (Figure 5a). At a soil depth of 15–30 cm, all models of N-cycling gene abundance were significantly correlated with important soil factors (Figure 5b).

## 4. Discussion

The security and sustainability of food production is one of the key challenges of the 21st century, and mixture and rotation are essential for soil fertility maintenance as conservation tillage practices. Our study found that complex cropping patterns promoted soil nutrients and prevented the further development of soil salinization (Table 1). This seems to be due to the effective increase of root secretions containing organic acids with the use of mixture and rotation, which in turn neutralized the $CO_3^{2-}$ and $HCO_3^-$ content on soil colloids [52]. In addition, plants could absorb soil salts because of their ability to absorb salt ions directly [53,54].

### 4.1. Complex Cropping Patterns Alter nifH Gene Expression

Soil nitrogen is critical and the lack of nitrogen will severely limit agricultural production [55]. In the biosphere, nitrogen can only be fixed naturally by nitrogen-fixing microorganisms [56]. Zhou et al. found that crop rotation significantly affected the community structure of *nifH* diazotrophs, especially the community composition, by comparing soil samples from uncultivated and rotation fields [57]. Our study showed that soil depth may be an important factor affecting *nifH* expression in rotation soils, specifically by decreasing in topsoil and increasing in deeper soils. Previous studies have shown that high soil $NH_4^+$ amounts can suppress the abundance of the *nifH* gene [58–60]. Unfortunately, we did not test the $NH_4^+$ content of the three cropping patterns. Therefore, we speculate that it may be caused by high plant N uptake and N loss from different plants in the rotation soils triggered by heavy summer rainfall [61]. Hao et al. [62] also showed by a meta-analysis that long-term crop diversification appears to lead to a decrease in the abundance of the *nifH* gene. Our study found that the lowest AP content was detected in surface rotation soils (Table 1). The decrease in the abundance of *nifH* in the soil also seems to be closely related to the reduction of AP content [63]. Furthermore, we noted that proteases were the most important factors driving the abundance of the *nifH* gene (Figure 4). Soil microorganisms are capable of secreting extracellular proteins to influence soil nutrient cycling, so changes in proteases that are sensitive to the external environment may affect soil microbial-mediated nutrient cycling [64,65]. Proteases could suppress the abundance of the *nifH* gene from rotation soil in 0-15 cm soil depth, even though the two do not show significant correlation (Figures 3d and 5a). This may be due to the fact that proteases provide bioavailable nitrogen by breaking down organic matter in the soil [66] and smaller organic matter molecules such as oligopeptides and amino acids [67]. In addition, Wang et al. also showed that the abundance of *nifH* may also be negatively correlated with bioavailable nitrogen in soil due to changes in the ecological niche, which further confirms our speculations [68]. The abundance of *nifH* increases with increasing soil depth. It appears to exhibit higher abundance for rotation soils in the 15–30 cm soil depth [69]. By comparing continuous and rotational cropping treatments, Zou et al. [70] showed that rotational cropping systems could significantly increase the abundance of the *nifH* gene at 0–20 cm soil depth in the black soils of Northeast China. This may be due to the perturbation of the multi-year crop rotation promoting an increase in the abundance of the *nifH* gene that was not as stable as in monoculture [71]. In addition, the abundance of *nifH* seems to be associated with soil $N_2$ fixation [72]. It has been shown that the abundance of *nifH* genes detected in Australian soils during the disturbance experiment was significantly and positively correlated with the rate of nitrogen-fixation processes, indicating the vigorous activity of bacterial populations [73]. By comparing the abundance of functional genes mediating different nitrogen-cycling processes, we found that the nitrogen-fixation efficiency of rotation and mixture soils may be higher than that of monoculture in deep soil (Figure 2a). The increase in soluble organic carbon promoted by the litter left in the soil by rotation and mixture can provide a sufficient carbon source for the enrichment of diazotrophic bacteria [72], which promotes the nitrogen-fixation efficiency of the rotation and mixture in the deep soil (Figure 2a). In addition, we noted that soil depth significantly influenced the nitrogen-fixation efficiency in complex cropping systems (Figure 2a). Deeper soils have higher soil temperatures than topsoil, which promotes the decomposition of soil organic matter and thus increases the efficiency of N fixation. Soil quality improvement can improve greenhouse-gas ($N_2O$) emissions, and the significant correlation between nitrate reduction and the abundance of *nifH* may be due to the relationship established by $N_2O$ emissions [74]. It has been established that $N_2O$ emissions appear to be positively correlated with nitrate-reductase activity, while a reduction in soil $N_2O$ emissions is able to increase the abundance of N-fixing microorganisms [74]. Soil nitrate reductase may have influenced $N_2O$ emissions and, in turn, the abundance of *nifH*. The increased abundance of *nifH* is supported by the possible higher availability of oxygen in the soil, as low oxygen conditions are also considered to be one of the controlling factors for microbial ni-

trogen fixation [75,76]. In addition, deeper soils have higher soil temperatures than topsoil, which promotes the decomposition of soil organic matter and thus increases the efficiency of N fixation.

*4.2. Cropping Patterns Ecological Niches Affect Nitrifying Bacteria Abundance*

Nitrification can be driven by AOB or AOA and is a central process in the N-cycling process [77]. Our study found that the abundance of AOA of monoculture was significantly higher than in rotation and mixture soils in surface soil, which seemed to be caused by differences in soil oxygen levels (Figure 1). The multi-year rotation of soils introduces different vegetation types of apoplastic material, which can act as cover crops that can play a role in increasing soil oxygen [78]. In addition, mixture soils have greater planting density, and more plant roots and an excess root system ensure sufficient soil pore space to store oxygen [79]. Interestingly, we noted a significant negative correlation between cellulase activity and AOA gene abundance (Figure 5a). Zhang et al. [80] reported that elevated $CO_2$ can lead to a decrease in soil-cellulase activity. This indicates that the link between AOA abundance and the presence of cellulose seems to be caused by changes in $CO_2$ due to changes in oxygen content. In contrast to the abundance of AOA, more complex cropping patterns appear to have led to an increase in the abundance of AOB in 0–15 cm soil depth (Figure 1). Although a study reported that crop rotation seems to have little effect on AOB abundance [81], for saline low-nutrient soils, the restoration of soil nutrients seems to provide more ecological niches for increased AOB abundance. Rotation and mixture will inevitably introduce more nutrient inputs than monoculture soils, and this will inevitably increase the abundance of AOB [82,83]. Many studies have confirmed that pH is a driving factor affecting the abundance of AOB [84–86]. Our study found that soil pH was significantly and negatively correlated with the abundance of AOB (Figure 5a). It was reported that AOA seems to prefer an acidic environment while AOB prefers an alkaline environment [87]. AOB is more sensitive to environmental changes and more susceptible to soil-salt stress than AOA [88]. Saline and alkaline stress reduced the number of AOB and reduced the preference of AOB for saline and alkaline environments [89]. Furthermore, in addition to soil acidity, the ecological niche specialization between AOA and AOB depends on the availability of nutrients from soil resources [90]. Soil proteases and β-glucosidases have been shown to be significantly correlated with soil-nutrient content [91], suggesting that AOB seems to be able to benefit soil nutrients better than AOA to ensure bacterial population growth. In addition, a significant negative correlation between EC and AOB abundance has also been reported, and EC may have an effect on AOB abundance [92]. Luo et al. [93] also reported a significant positive correlation between soil AP content and AOB abundance. In deeper soils, we observed significantly higher abundances of both AOA and AOB in rotation than in monoculture and mixture soils (Figure 1). The changes in AOA and AOB abundance were consistent, showing that AOA also plays an important role in soil nitrification in deep soils [94]. Crop rotation can better increase the carbon and nitrogen content of the soil and maintain soil quality as well as nutrient balance [95,96]. It has been confirmed that nitrate reductase is closely related to the AOA community [97].

We noted that the mixture possessed higher *nxrB* abundance than monoculture and rotation at a soil depth of 0–15 cm (Figure 1). It is reported that *nxrB* is a functional gene of the *Nitrospira* bacterial populations [98] and mixture soils are richer in root secretions and thus recruit *Nitrospira* [99]. We also found that the abundance of *nxrB* appeared to be significantly higher in monoculture than in a mixture at a soil depth of 15–30 cm (Figure 1). Zhang et al. [100] found that a conventional monocrop pattern could increase the abundance of *Nitrospira*, which may explain the higher abundance of *nxrB* in monoculture soils. *Nitrospira* is a diverse group of nitrite-oxidizing bacteria (NOB) and among the environmentally most widespread nitrifiers [101] and it responded significantly to the gradient of environmental nutrients, which included soil TP content [102]. In addition, Yu et al. [103] tested nine classes of *Nitrospira* and three of them showed a significant positive correlation with soil TP, which would suggest that a significant positive correlation

between *nxrB* and TP is possible. Hu et al. found that the AOA/AOB ratio decreased with increasing pH by gaining insight into the ecological characteristics of AOB in 65 soils collected from various soil and ecosystem types [104]. This contrasts with our findings that AOA microorganisms seem to remain active in saline soils and that AOA and AOB do not compete. In addition, the higher soil nutrient content of rotation and mixture promoted the increase in AOB abundance, resulting in a lower AOA/AOB ratio [105]. Szukics et al. [106] found that AOA abundance can also be favored by high pH through the study of eight mountain grasslands in Austria, France, and the UK. In addition to this, we observed a significant effect of soil depth on AOA/AOB (Figure 2). AOA has the advantage of being more numerous in the amoA gene pool relative to AOB, and as AOA decreases with depth leading to the same trend in the AOA/AOB ratio [107].

*4.3. Mixture and Rotation Reduce N Losses by Reducing narG and nosZ Abundance*

In this study, we observed that the abundance of *narG* and *nosZ* was significantly higher in monoculture than in rotation and mixture soils in a soil depth of 0–15 cm (Figure 1). The decrease in the abundance of *narG* may be related to the shallow groundwater of the soil. Zhang et al. found that the abundance of *narG* genes decreases as the water table decreases and the soil profile continues to dry [108]. Crop rotation soils have more plant litter in the surface soil to play a certain role in water absorption [109]. In addition, more roots in mixture soils also led to increased water demand [110]. Soil pH appears to be an important factor affecting *nosZ* abundance, and the abundance of *nosZ* was inhibited by low pH and increased with increasing pH [111]. The present study confirms this idea, with lower pH values in rotation and mixture than in monoculture soils (Figure 3d and Table 1). In addition, an increase in the denitrifying bacteria abundance and a significant decrease in urease activity have been reported, which is consistent with the results of the study (Figure 3d). The abundance of *narG*, *norB* and nosZ genes were significantly higher in rotation than in monoculture and mixture soils in 15–30 cm soil depth (Figure 1) Crop rotation can increase the rate of denitrification, which also increases when a mixture of crop residues is added to the soil [112]. It seems that the difference in oxygen levels due to soil depth provides a better environment for denitrifying bacterial populations to survive, and residual apoplast is less likely to decompose and meet nutrient requirements. In addition, we also observed that the abundance of *nosZ* in the mixture was significantly higher than in monoculture in deep soils (Figure 1). Castellano et al. demonstrated that mixed crops of both legumes and non-legumes significantly increased *nosZ* abundance and significantly reduced $N_2O$ emissions by mixing two types of plants, legumes, and non-legumes [113]. This may be due to the influence of soil moisture. It has been reported that *nosZ* gene abundance is increased by reducing soil moisture [114]. More roots in the mixture reduce soil water retention compared to monoculture soils in the deep soil. Notably, we found that the *norB*/*nosZ* ratio in the mixture soils was significantly higher than the monoculture soils in surface soil (Figure 2d). This suggests that $N_2O$ emissions in monoculture may be greater than in mixture soils. A tighter link between $N_2O$-reducers and mixture-composition compared to microbes involved in $N_2O$ production, which may be of agronomic relevance as microbes containing the *nosZ* gene are the only known biological sink for $N_2O$ emissions [115]. This indicates that the abundance of *nosZ* is closely related to $N_2O$ emissions, and the abundance of *nosZ* is significantly higher in monoculture than in mixture soils in 0–15 cm soil depth (Figure 1). Unlike the topsoil, the *norB*/*nosZ* ratio was significantly lower in the mixture than in the rotation and monoculture soils in 15–30 cm (Figure 2d). This may be due to changes in microbial community structure as well as changes in *nosZ* community diversity. Different mixed-crop species affect the abundance of *nosZ* and *nosZ* communities [116]. This is due to differences in the composition of plant root exudates and root morphology, which determine the degree of competition between plant and microbial communities for available soil N [117,118]. In addition, our study found a significant correlation of the *norB*/*nosZ* ratio with soil EC and protease activity (Figure 3e). Adviento et al. found that the maximum $N_2O$ loss occurred at the highest EC

level by exploring the effect of electrical conductivity (EC) on $N_2O$ production in five soils under intensive cultivation [119]. It has also been confirmed that soil $N_2O$ emissions are significantly correlated with soil protease activity [120], which remains consistent with the results of this study (Figure 3e).

## 5. Conclusions

The introduction of a mixture of oats and tall wheatgrass, along with rotation, has been found to promote soil desalination and nutrient restoration, particularly soil enzyme activity, in saline soils. The abundance of N-cycling is strongly influenced by cropping patterns and soil depths, with rotation decreasing *nifH* abundance in surface soils and increasing *nifH* abundance, AOA, and AOB abundance in deep soils. Crop diversity is crucial, resulting in the highest *nxrB* abundance in mixture surface soils. However, complex cropping patterns decrease AOA abundance and increase AOB abundance in surface soils. They also reduce *narG* and *nosZ* abundance in surface soils while increasing denitrification in deep soils. Mixtures may promote $N_2O$ emissions from surface soils and suppress $N_2O$ emissions in deep soils. Changes in soil chemistry and enzyme activity play a vital role in the genetic variation of N-cycling functions in saline soils. Further investigation into changes in microbial communities of different N-cycling genes is necessary to better understand the mechanisms of genetic variation in nitrogen-cycling processes in saline soils.

**Supplementary Materials:** The following supporting information can be downloaded at: https://www.mdpi.com/article/10.3390/su15086592/s1, Table S1: Summary of genes investigated, the enzymes they encode, and their function in the nitrogen cycle; Table S2: PCR primers used for quantitative PCR and reaction conditions. References [121–126] are cited in the Supplementary Materials.

**Author Contributions:** Conceptualization, J.D., M.S. and X.L.; methodology, J.D. and B.L.; data processing, J.D. and M.S.; literature review, B.L. and J.D.; writing—original draft preparation, B.L. and J.D.; writing—review and editing, X.L.; supervision, M.S. and X.L.; funding acquisition, J.D. All authors have read and agreed to the published version of the manuscript.

**Funding:** This study received financial support from the Natural Science Foundation of Heilongjiang Province (Grant No. LH2021D014).

**Institutional Review Board Statement:** Not applicable.

**Informed Consent Statement:** Not applicable.

**Data Availability Statement:** The datasets generated during and/or analyzed during the current study are available from the corresponding author upon reasonable request.

**Conflicts of Interest:** The authors declare no conflict of interest.

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
