# Peer review of "Different Cropping Patterns to Restore Saline-Alkali Soils in Northeast China Affect the Abundance of Functional Genes in the Soil Nitrogen Cycle"

_sustainability, doi:10.3390/su15086592_

Round 1
Reviewer 1 Report
The manuscript by Ding et al. describes the results of a study that analyzes the effect of monoculture, crop rotation and mixed plantings on the abundance and distribution of microbial populations (ammonia-oxidizing bacteria and ammonia-oxidizing archaea) as well as genes encoding nitrogen metabolic enzymes (nifH, nxrB, narG, norB, nosZ). These were measured by quantitative PCR. Other soil parameters are also measured, including pH, soil organic matter, total phosphate, total nitrogen, available phosphate, available potassium, electric conductivity, as well as the soil enzymes urease, nitrate reductase, cellulase, beta-glucosidase and protease.
Details:
Line 100: “the annual effective temperature is 2500 – 2700oC” this sounds very high to me
Table 1: It is not clear whether enzyme amounts or enzyme activities were measured. (The methods were not described and reference 46 was not available to me). Anyway, the enzyme units do not make sense. The only one that would make sense is the unit for the betaglucosidase which is given in microgram per g and hour. This could be the activity of the enzyme describing how much substrate was converted per gram soil and per time unit. All the other measurements are in mg which denotes a mass. The protease is given in microgram per milliliter per meter which does not make sense, maybe the authors meant minute?
Line 229: “ration” should be “ratio”
Lines 317/318: it is not clear why proteases should suppress the abundance of the nifH gene. Why?
Lines 342/343: what should be the cause of lower N2O emission being able to increase the abundance of N2-fixing microorganisms?
Lines 376, 387, 392, 422: “colony” -usually you only talk about bacterial colonies, when they grow on plates in the laboratory. Bacteria rarely grow in colonies in nature.
Line 392: Nitrospira to be rhizobial – I do not think so. Nitrospira is not Rhizobium and Rhizobium is not Nitrospira
Reviewer 2 Report
The authors of the paper entitled "The Process of Cropping Patterns to Restore Saline-alkali Soils in Northeast China Affects the Abundance of Functional Genes in the Soil Nitrogen Cycle" evaluate whether the abundance of N-cycle genes in soil responded to changes in the crop soil systems and depths. The research is very interesting as saline-alkaline soils are a problem affecting millions of hectares worldwide.
I find the paper well structured considering the analyzes carried out to verify the hypotheses of the authors.
I consider it interesting for readers.
Further investigations about changes in microbial communities of several functional nitrogen cycle genes are needed to better understand the mechanisms of changes in nitrogen cycle processes in saline soils.
Therefore, your research can be considered an interesting
starting point and needs some revisions before it is ready for publication.
Firstly, it is suggested that authors rewrite the title. It needs to be more concise and attract more readers' attention.
Details should be included in the "materials and methods" section. Information relating to the soil texture and the type of fertilizers used could help the reader to better understand your research.
The authors using a saline-alkaline experimental soil. What classification are they referring to? Has the SAR been calculated? If yes, it could be inserted in the text.
Also, the results should be rewritten.
This part is very repetitive. It is advisable to vary the construction of the sentences to improve the quality of the manuscript.
The authors could improve section 5. This section is too long to be conclusions. You should focus on how your hypotheses were tested and goals achieved.
Finally, it is suggested to update some bibliographic references.
Round 2
Reviewer 1 Report
Title:
“Process of..” does not make much sense. I suggest the following title: “Different cropping patterns to restore saline-alkali soils in Northeast China affect the abundance of functional genes in the soil nitrogen cycle”
Table 1:
The units for the enzyme activities are starting to make more sense, however, for the urease and nitrate reductase activities the “1” before day should be omitted (it is understood that it is per "1 day"; you do not write per "1 hour" either, but just "hour "for the beta glucosidase and protease activities). But, most importantly, the cellulase activity should not be expressed per 3 days. It also has to be expressed per one day. If the value is getting too low in milligrams, use micrograms.
Lines 142 and 143: amoA should be italicized
Lines 255 to 266: Why are there so many semicolons highlighted in green? No single semicolons should be in this paragraph.
Line 308: Add “crop” before “rotation” to clarify
Line 356: the word soil “microcosm” comes out of the nowhere and it is used only once. Which microcosms are meant? Are they part of this study or part of the references 75 and 76?
Lines 384/385: this is only a half sentence, something is missing there
